# Acoustic delivery of indocyanine green via biosynthetic gas vesicles for tumor photothermal therapy

Jiaqi Zhang[1,2◉], Licong Huang[3◉], Shuhui Wang[4◉], Jingwen Ding[1], Yuping Yang[5], Chenhui Li[4], Ping Zhao[1]*, Qian Li[4]*, Fei Yan[6]*

**1** Department of Ultrasound, The First Affiliated Hospital, Guangzhou University of Chinese Medicine, Guangzhou, China, **2** Department of Ultrasound, Xijing Hospital, The Fourth Military Medical University, Xi'an, China, **3** Department of Ultrasound, The Third Affiliated Hospital of Sun Yat-Sen University, Guangzhou, Guangdong, China, **4** Department of Ultrasound, The Affiliated Cancer Hospital of Zhengzhou University & Henan Cancer Hospital, Zhengzhou, China, **5** Department of Ultrasound, The Affiliated Hospital of Guangdong Medical University, Zhanjiang, China, **6** State Key Laboratory of Quantitative Synthetic Biology, Shenzhen Institute of Synthetic Biology, Shenzhen Institutes of Advanced Technology, Chinese Academy of Sciences, Shenzhen, China

◉ These authors contributed equally to this work.
* pingzhao1499@126.com (PZ); zlyyliqian1018@zzu.edu.cn (QL); fei.yan@siat.ac.cn (FY)

## Abstract

Photothermal therapy (PTT) serves as a complementary strategy to conventional cancer treatments. Indocyanine green (ICG) is the only U.S. Food and Drug Administration (FDA)-approved photothermal agent. However, its clinical application is hindered by poor stability, short blood half-life, and lack of tumor targeting. Herein, we developed biosynthetic gas vesicles (GVs) covalently conjugated with ICG (ICG-GVs) for remotely controlled, visibly acoustic delivery of ICG to tumors in a subcutaneous xenograft model of MB49 murine bladder cancer in C57BL/6 mice. The resulting ICG-GVs exhibit uniform morphology (~200 nm) with an ICG loading rate of 58%, good colloidal stability, and enable trimodal imaging (ultrasound, near-infrared fluorescence, and photoacoustic) for real-time visualization of delivery. Pharmacokinetic analysis revealed that ICG-GVs significantly prolong ICG circulation half-life and increase AUC. Ultrasound-triggered GV cavitation enhanced intratumoral ICG delivery, achieving tumor temperatures >60 °C upon laser irradiation, leading to complete tumor regression and prolonged survival without detectable toxicity. This study provides a clinically translatable strategy for precise and effective ICG-based PTT.

## 1. Introduction

Photothermal therapy (PTT) is one of the most rapidly evolving cancer treatment modalities in recent years. It uses light to induce local temperature increase in tumors administrated various photothermal agents. Due to its distinct mechanisms of action, PTT has attracted widespread attentions for its advantages in the noninvasive

**Data availability statement:** All relevant data are within the paper and its Supporting information files. All underlying data for Figs 2D–2G, 3D–3F, 4D, 4H, 5B–5D, 5F, S2, S3, S4, S5A–S5H, and S6A–S6D are provided in S1 Data.

**Funding:** FY was supported by the National Key R&D Program of China (No. 2025YFA0922100), the Shenzhen Medical Research Fund (No. B2402006, A2502037), and the Leading Talents of the Guangdong Special Support Program (No. 2024TX08A098). SHW acknowledges funding from the Henan provincial Medical Science and Technology Research Project (No. SBGJ202502034), and the Natural Science Foundation of Henan (No. 252300420567). The funders had no role in study design, data collection and analysis, decision to publish, or preparation of the manuscript.

**Competing interests:** The authors have declared that no competing interests exist.

**Abbreviations:** ANOVA, analysis of variance; CEUS, contrast enhancement ultrasound; FBS, fetal bovine serum; FDA, Food and Drug Administration; GVs, gas vesicles; H&E, hematoxylin and eosin; ICG, Indocyanine green; ICG-GVs, ICG-coating GVs; NIRF, near-infrared fluorescence; PA, photoacoustic; PI, propidium iodide; PTT, photothermal therapy; SD, standard deviation; SEM, standard error of the mean; TEM, transmission electron microscope.

process, few side effects, and high efficiency of tumor ablation. Also, PTT has been shown to overcome chemotherapy resistance, making it possible combine with other treatment modalities [1]. Furthermore, advances in various targeted delivery strategies of photothermal agents improve its spatiotemporal controllability of PTT, greatly increasing its safety and reducing off-target toxicities. Nowadays, numerous photothermal agents, such as gold nanocomposites, carbon nanomaterials, sulfide nanocomposites, and black phosphorus, have been developed, but their further clinical applications are hindered by the potential toxicity and long-term retention [2–5]. Therefore, it is desirable to develop novel photothermal agents on the basis of clinically licensed agents. Indocyanine green (ICG), the only near-infrared fluorescence (NIRF) imaging dye approved by the U.S. Food and Drug Administration (FDA), has been clinically used for photoacoustic (PA) imaging and PTT [6]. However, the poor water stability and short *in vivo* half-life of ICG impair its tumor-targeted accumulation and its therapeutic effect. Evidence has demonstrated that 97% of ICG will be rapidly removed from the blood only 20 min after intravenous injection of ICG [7]. There is an ongoing effort to develop variants of ICG with longer blood circulation times and tumor accumulation, some of them have proceed into clinical trials [8–10]. Moreover, the poor intratumoral delivery of ICG is another issue, making it difficult to achieve satisfactory therapeutic effect in PTT [11]. Thus, it is very important to develop novel strategy to improve the tumor delivery efficiency of ICG for enhancing the efficacy of PTT.

Ultrasound-targeted microbubble destruction is a novel technology used for local drug delivery into certain special tissues, such as tumors, heart, brain, etc. Thanks to the advantages of ultrasound in high tissue penetration depth and focusability, the sound wave can be positioned into the tumor, inducing the systemically administrated microbubbles to generate oscillation and to produce a series of acoustic cavitation effects, including strong shear forces, microjets, microstreaming, etc. [12–14]. These acoustic cavitation effects would perforate the vascular endothelial cells or enlarge the gap of endothelial cells, resulting in the transvascular delivery of drugs in the cavitation site. Also, drug-loaded microbubbles can be fabricated through coating therapeutic agents on the surface of microbubbles, encapsulating them into the shell or inner cavity of microbubbles, which further improves the drug delivery efficiency. However, traditional microbubles have micro-scale particle size, which confines them within blood vessel. It results in the cavitation effects mainly occur on vascular endothelial cells, limiting the drug delivery since these drugs have to overcome multiply cellular barrier to enter the tumor cells.

Recently, gas vesicles (GVs), a kind of novel nanobubbles from buoyant microbes are proven to have excellent ultrasound imaging and cavitation performance [15–17]. Importantly, they have approximately 200 nm particle size, making them able to easily transport out of tumor blood vessel via EPR effect and contact tumor cells. Upon receiving acoustic irradiation, they would generate the cavitation effects and directly deliver drug into the tumor cells. Taking the great advantages of GVs into account, herein, we developed a novel ICG-coating GVs (ICG-GVs) for enhancing and visualizing the ICG delivery process, and improving its anti-tumor efficacy of PTT.

## 2. Materials and methods

### 2.1 Preparation of ICG-GVs

Bacterial culture and GV isolation were carried out in accordance with our previous report and Shapiro and colleagues [12,15,16,18]. Briefly, *Halobacterium NRC-1* (Halo) was cultured for two weeks in ATCC medium at 37 °C with shaking at 220 rpm. These bacteria were then transferred to a separating funnel and left at room temperature for one week. After floating, the bacterial cells were separated and lysed with TMC buffer, followed by centrifugation at 300$g$ for 4 h. The milky GVs were isolated and stored at 4 °C. The ICG solution in PBS (pH = 7.4) was added to 1 mL of GVs (OD500 = 3.0) solution. The mixture was then incubated at 4 °C for 3 h. The mixture was then purified by centrifugation four times. The resulting mixture was added to an ultrafiltration tube (2 mL) and centrifuged at 300$g$.

### 2.2 Characterization of GVs and ICG-GVs

The particle size and surface zeta potential were determined by zeta analyzer (Malvern Nano ZS Zetasizer). The UV-absorption spectrum was characterized as follows: 100 μL of ICGs, GVs or ICG-GVs solution were transferred to a 96-well plate and measured by UV spectrophotometer. The concentration of the prepared ICG-GVs solution was quantified using the absorbance at 500 nm wavelength according to the method of Shapiro and colleagues [12,18]. The morphologic examination was further performed by transmission electron microscope (TEM). For comparing the photostability between ICG-GVs and free ICG, the samples were irradiated with 808 nm laser for 3 min, followed by cooling to room temperature for 10 min. This on-off cycle was repeated for five times and then the morphologies of irradiated ICG-GVs were examined by TEM.

### 2.3 *In vitro* multimodal imaging

The *in vitro* contrast enhancement ultrasound (CEUS) imaging of ICG-GVs was performed on a phantom of agarose (2% w/v). ICG-GVs at different concentrations ($OD_{500}$ = 2.2, 2.4, 2.6, 2.8, or 3.0) were added into the phantom holes. Imaging was performed using ultrasound system (Mindray Resona 7, Mindray, China) equipped with the L11-3U line array transducer (3.0–11.0 MHz). The parameters were kept as follows: acoustic power: 5.13%, mechanical index: 0.149, contrast gain: 65 db. The PA imaging was performed by system (VevoLAZR, VisualSonics, USA). ICG-GVs ($OD_{500}$ = 3.0) at different ICG linkage concentrations (10, 20, 30, or 40 μg/ml, $OD_{500}$ = 3.0) were added into the phantom holes and imaged by VevoLAZR.

For determining the photothermal effect, ICG-GVs ($OD_{500}$ = 3.0) with different ICG linkage concentrations (10, 20, 30, or 40 μg/ml) were added into the well of 96-well plate. NIR laser (808 nm, 0.8 W/cm$^2$) was used to irradiate the samples for 5 min. The changes of temperature were obtained by infrared imaging camera (Ti27, Fluke, USA).

### 2.4 Cell culture and animal model

MB49 murine bladder cancer cells were cultured in high-glucose DMEM containing 10% fetal bovine serum (FBS), 100 U/ml penicillin, and 100 μg/ml streptomycin at 37 °C. Female C57BL/6 mice (4–6 weeks old, 15–20 g) were maintained under standard environmental conditions, and received care in accordance with the Guidance Suggestions for the Care and Use of Laboratory Animals. The procedures were approved by Shenzhen Institutes of Advanced Technology, Chinese Academy of Sciences Animal Care and Use Committee (Ethics No. SIAT-IACUC-20211202-YGS-YXZX-YF-A1693-01). Animal research was performed according to ARRIVE guidelines and the National Institutes of Health guide for the care and use of Laboratory animals (NIH Publications No. 8023, revised 1978). To establish the tumor model, 4 × 10$^6$ MB49 cells suspended in 100 μL PBS were injected into the right groin area of mice.

### 2.5 *In vitro* photothermal efficiency

MB49 cells were seeded in 96-well plates (5 × 10$^3$ cells/well) for 24 h. They were divided into six groups, including the control group, ICG-GVs group, ICG-GVs + US group, ICG-GVs + Laser group, US+Laser group, ICG-GVs + US + Laser group.

In brief, 20 μL ICG-GVs ($OD_{500}$ = 3.0; ICG: 40 μg/mL) were added to 100 μL PBS and then pulsed ultrasound power were performed for ICG-GVs + US group, US+Laser group, ICG-GVs + US + Laser group for 1 min. Laser irradiation at $0.75 W/cm^2$ was treated for these cells of ICG-GVs + Laser group, ICG-GVs + US + Laser group for 5 min. After 4 h incubation, cell viability was determined by the Cell Counting Kit-8 kit (Dojindo, Japan) by measuring the absorbance at 450 nm using a multimode plate reader (Synergy 4, BioTek, VT, USA). To visually observe the photothermal therapeutic efficacy, these treated MB49 cells were seeded onto 96-well plate. After another 24 h incubation, cells were washed with PBS and then stained with Calcein-AM/propidium iodide (PI) double staining Kit, followed by observation under microscope.

## 2.6 *In vivo* multi-imaging guided PTT assay

The multimodal imaging-guided PTT were performed when tumor volumes of mice reached to about 120 mm³. The tumor-bearing mice were randomly divided into six groups, including the control group, ICG-GVs group, ICG-GVs + US group, ICG-GVs + Laser group, US + Laser group, ICG-GVs + US + Laser group. Eight mice were used in each group. ICG-GVs (100 μL at $OD_{500}$ = 3.0) or PBS were injected into mice via the tail vein. The ICG-GVs were prepared with an ICG concentration of 40 μg/mL ($OD_{500}$ = 3.0). And ultrasound imaging was performed using a 3.0–11.0 MHz line array transducer equipped by Mindray Resona 7. The acoustic output parameters were set within the safety limits recommended for diagnostic and therapeutic applications [19,20]. When CEUS imaging showed that ICG-GVs perfused the whole tumor, acoustic power bursts (−3 dB) was performed every 10s for ICG-GVs + US and ICG-GVs + US + Laser groups until the ICG-GVs disappeared. At 4 h post-injection, the tumors of mice in the ICG-GVs + Laser group and ICG-GVs + US + Laser group were irradiated by a diode laser ($\lambda$ = 808 nm, 0.75 W) for 20 min. The maximum temperature of the tumor surface was determined using an FTIR thermal camera (E4, FLIR Systems). The tumor volume and mouse weight were measured every two days. Tumor volume was calculated as follows: tumor volume = (tumor length × tumor width²)/2. The study was continued until the average tumor size reached 2,000 mm³ or until a mouse died, and then the survival curves were recorded. Tumor apoptosis was also assessed by TUNEL assay according to the product instruction.

## 2.7 Biocompatibility of ICG-GVs

The cytotoxicity of ICG-GVs was evaluated by the CCK8 assay. Briefly, MB49 cells at 5,000 cells/well were inoculated in 96-well plates, and the culture medium was replaced with fresh culture medium after 24 h. Twenty microliters of ICG-GVs (with ICG concentrations at 10, 20, 30, or 40 μg/mL) was added to each well. Six sub-wells were set up for each concentration, and the cells which did not receive with ICG-GVs were set up as the control group, and the cell-free group was set up as the blank group. The cells were incubated for 12 h in the $CO_2$ incubator. After that, the medium was renewed, 10 μL of CCK8 solution was added, and the cells were further incubated in the $CO_2$ incubator for 1 h. The OD value of each well at 450 nm was measured by multifunctional enzyme marker.

## 2.8 Histology and immunohistochemistry

After the tumor treatment experiments, mice (*n* = 3) were randomly executed, and tumors, kidneys, livers, lungs, spleens, and hearts were removed, followed by fixing with 10% formalin. Tissue samples were embedded in paraffin and then were cut (8 μm thick). Paraffin sections were stained with hematoxylin and eosin (H&E) reagent to assess any tissue toxicity.

## 2.9 Statistical analysis

Statistical analyses were performed using GraphPad Prism 9.5 (GraphPad Software, USA) and SPSS v22.0 (SPSS, USA). The normality of continuous data was evaluated by the Shapiro-Wilk test or Kolmogorov–Smirnov test. For comparisons between two groups, unpaired two-tailed Student *t* test was used for normally distributed data, while Mann–Whitney *U* test was applied for non-normally distributed data. For comparisons among multiple groups, one-way analysis of

variance (ANOVA) followed by Tukey's post-hoc test was used for normally distributed data, and Kruskal-Wallis test with Dunn's post-hoc test was used for non-normally distributed data. For tumor growth curves involving repeated measurements over time, two-way repeated measures ANOVA was performed followed by Sidak's multiple comparisons test at each time point. Survival curves were analyzed using the Kaplan–Meier method with log-rank test. All data are presented as mean ± standard deviation (SD) or mean ± standard error of the mean (SEM) as indicated in the figure legends. A $P$ value < 0.05 was considered statistically significant.

## 3. Results

### 3.1 Synthesis and characterizations of ICG-GVs

The GVs were biosynthesized by *Halobacterium NRC-1* and then isolated and purified as shown in S1 Fig. The ICG-GVs were fabricated through conjugating ICG-NHS onto the surface amino group of GVs via amide bond formation and the ICG loading rate was 58% (Fig 1A). The resulting ICG-GVs appeared green due to the presence of ICG and showed obvious NIRF signal (Fig 1B). The TEM images revealed that GVs and ICG-GVs have a uniform rugby-shaped morphology (Fig 1C), showing that the conjugation of ICG did not damage the structure of GVs. Surface binding with ICG did not alter GV morphology but elevated their zeta potential to −22.0 ± 2.0 mV, and slightly increased the particle size of 264.0 ± 2.6 nm (Fig 1D and 1E). UV-VIS spectroscopy revealed the characteristic absorption peaks of ICG for ICG-GVs, confirming the successful ICG conjugation onto the GVs (Fig 1F). The resulting ICG-GVs could be stable in PBS for at least 7 days (S2 Fig). *In vitro* studies indicated that the characteristic absorption peaks of the solution containing ICG-GVs elevated gradually with the increase of ICG concentration (Fig 1G).

### 3.2 *In vitro* and *in vivo* imaging performance of ICG-GVs

Since GVs can be used as ultrasound contrast agents and ICG has NIRF imaging and PA imaging capabilities, we wonder whether ICG-GVs can be detected by ultrasound imaging system, NIRF imaging system, and PA imaging system. To test it, we added different concentrations of ICG-GVs into the agar phantom holes and used the equivalent GVs or ICG as the controls, followed by ultrasound or optical imaging. To quantitatively evaluate the multimodal imaging performance of ICG-GVs, we analyzed the *in vitro* imaging data. As shown in Fig 2D, the CEUS signal intensity increased in a concentration-dependent manner as ICG-GVs concentration increased from $OD_{500} = 2.2$ to 3.0, with no significant difference between ICG-GVs and unmodified GVs, indicating that ICG conjugation did not compromise the ultrasound imaging capability of GVs. Fig 2E demonstrates that the PA) signal intensity of ICG-GVs increased linearly with increasing ICG concentration (10–40 µg/mL), confirming its excellent PA imaging performance. Correspondingly, quantitative analysis of NIRF imaging (Fig 2F) showed that the fluorescence intensity of ICG-GVs gradually increased with ICG concentration, consistent with the fluorescence characteristics of free ICG. These results demonstrate that ICG-GVs retain both the ultrasound imaging capability of GVs and the optical imaging capability of ICG, achieving integrated multimodal imaging. Next, we further detected the imaging performance of ICG-GVs in tumor-bearing mice ($OD_{500} = 3.0$, ICG = 40 µg/mL). Strong CEUS signals can be observed in the tumors of mice received with ICG-GVs or GVs. There are comparable CEUS signal intensities in the tumors between ICG-GVs and GVs (Fig 2G). Interestingly, the infusion process of ICG-GVs can clearly visualized and tracked in the tumors. Notably, the NIRF signals of ICG-GVs in the tumor were stronger than that of ICG group at 30 min post-injection, suggesting that ICG-GVs may extend the halftime in the circulation and improve their tumor accumulation capability due to their nanoparticle size (Fig 2H).

### 3.3 Ultrasound irradiation enhanced ICG delivery into tumor cells

In order to examine whether acoustic cavitation enhances the delivery of ICG into the tumor cells, we incubated the ICG-GVs with MB49 tumor cells and followed by acoustic bursts every 10 s intervals for 1 min. Significantly more ICG

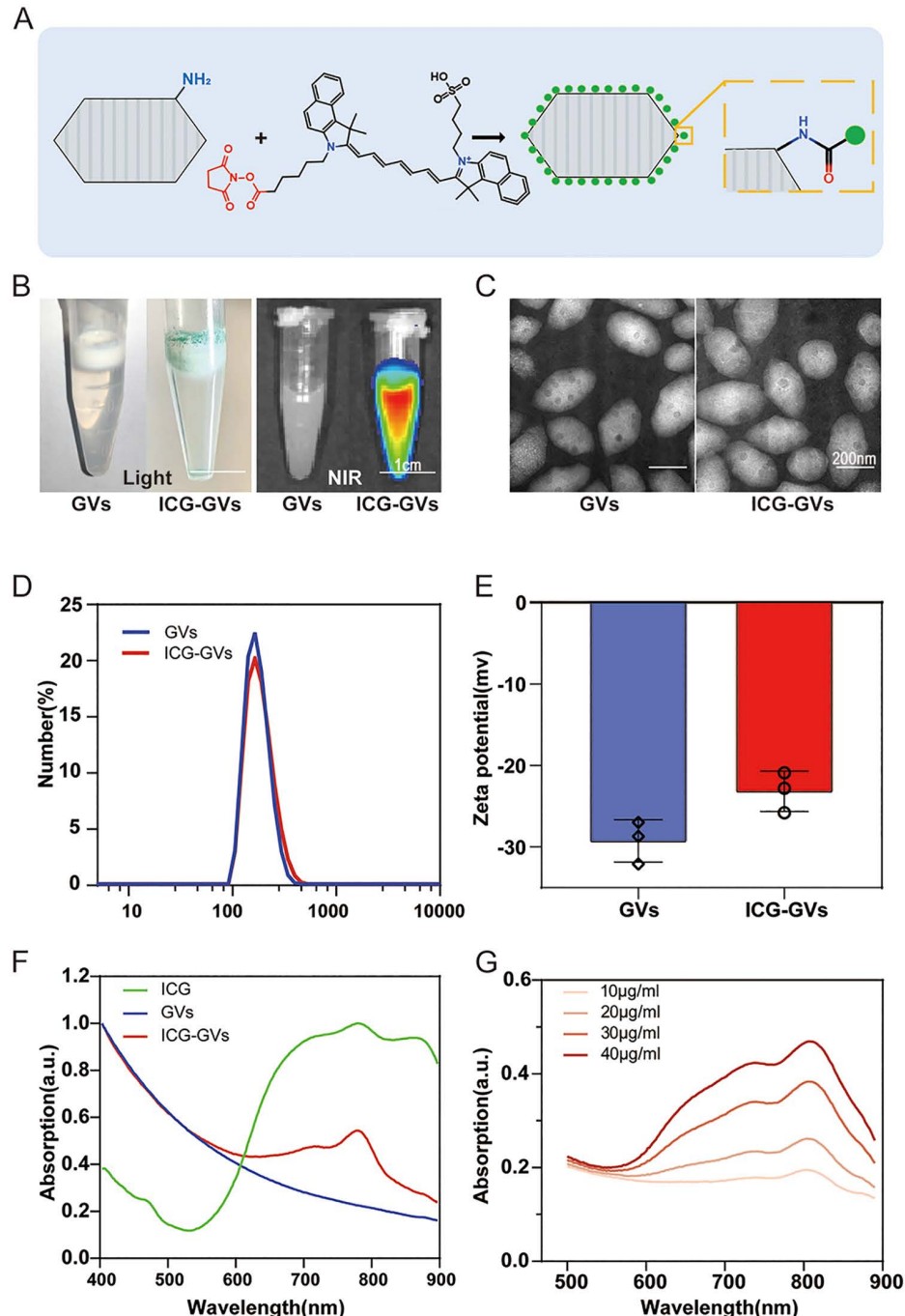

**Fig 1. Synthesis and Characterizations of ICG-GVs. (A)** GVs isolated from Halobacterium salinarum NRC-1 were coated with ICG through amide linkage. **(B)** ICG-GVs appeared green due to ICG-coated and showed near-infrared fluorescence signal. **(C)** TEM micrographs showing the morphology of GVs and ICG-GVs. Scale bar = 200 nm. **(D)** Size distribution and (E) zeta potential of GVs and ICG-GVs. Data are mean ± SD. Each dot represents an individual measurement (*n* = 3 per group). **(F)** UV-VIS spectrum of ICG-GVs and ICG and GVs. **(G)** UV-VIS spectrum of ICG-GVs coated with different ICG concentration. The underlying numerical data for this figure can be found in S1 Data.

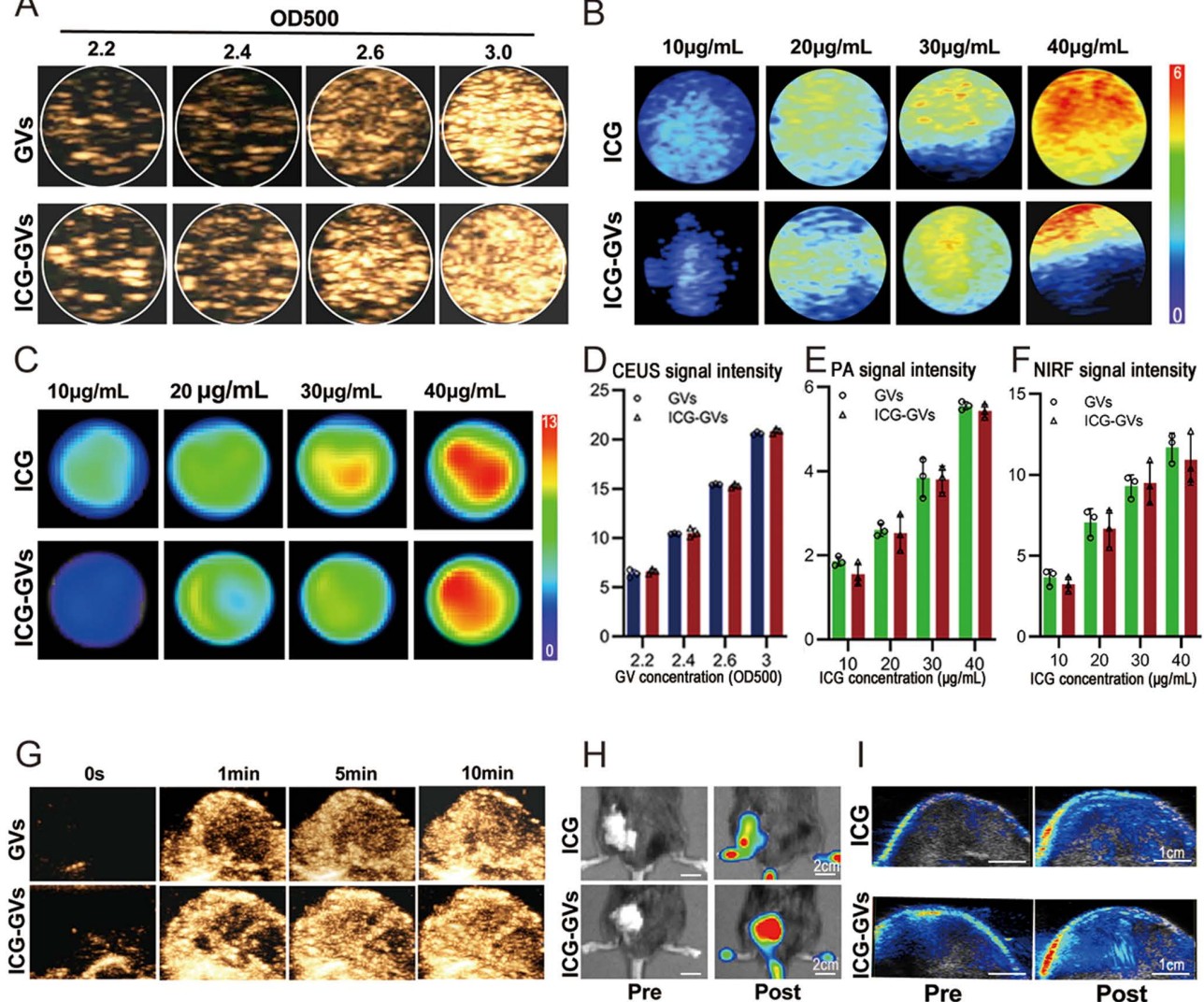

**Fig 2. *In vitro* and *in vivo* imaging performance of ICG-GVs. (A)** Ultrasound image in nonlinear contrast mode, (B) photoacoustic image, and (C) near-infrared fluorescence image of GVs and ICG-GVs *in vitro*. **(D)** Quantitative analysis of CEUS signal intensity corresponding to **(A)**. Data are mean ± SD (*n* = 3 per concentration). **(E)** Quantitative analysis of PA signal intensity corresponding to **(B)**. Data are mean ± SD (*n* = 3 per concentration). **(F)** Quantitative analysis of NIRF signal intensity corresponding to **(C)**. Data are mean ± SD (*n* = 3 per concentration). **(G)** Ultrasound image in nonlinear contrast mode, **(H)** near-infrared fluorescence image, and **(I)** photoacoustic image of GVs and ICG-GVs in tumor-bearing mice. **(G)** Ultrasound image in nonlinear contrast mode, **(H)** near-infrared fluorescence image, and **(I)** photoacoustic image of GVs and ICG-GVs in tumor-bearing mice. The underlying numerical data for this figure can be found in S1 Data.

fluorescence could be observed in the irradiated tumor cells relative to these non-irradiated tumor cells, showing the acoustic cavitation could improve the intracellular delivery of ICG into tumor cells (Fig 3B). After washing out the free ICG-GVs, we performed the *in vitro* PTT to these tumor cells by using of laser irradiation. The *in vitro* PTT cytotoxicity were determined by the calcein AM/PI staining assay. Fig 3C showed that laser irradiation of tumor cells that did not receive acoustic irradiation did not display evident cytotoxicity, attributing to the absence of ICG in these cells. By contrast, laser irradiation produced apparent cytotoxicity to these tumor cells that received acoustic irradiation. Quantitative analysis revealed that the ICG-GVs + US + Laser group achieved significantly higher tumor cell killing compared to the

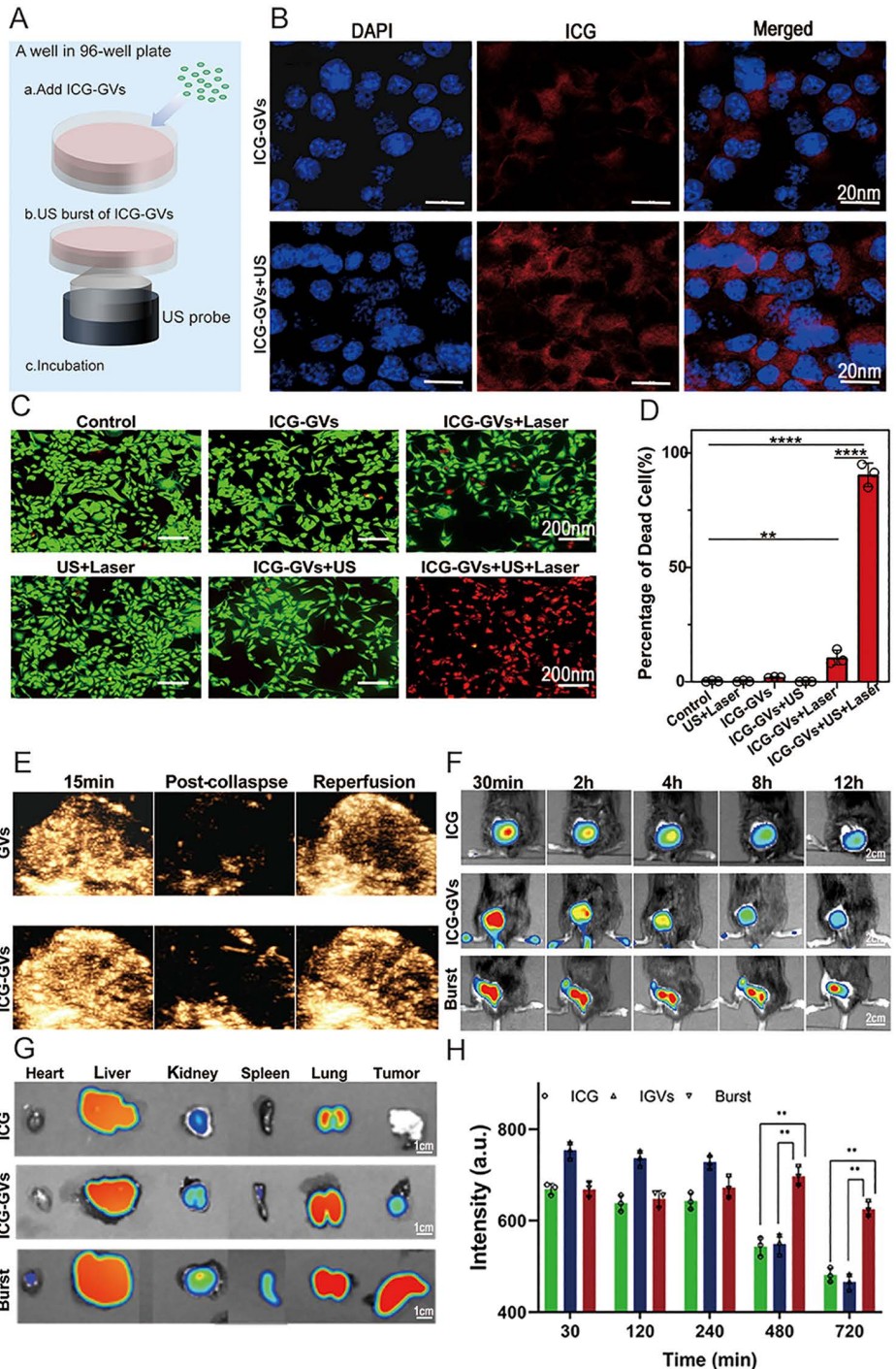

**Fig 3. Ultrasound irradiation enhanced ICG delivery into tumor cells. (A)** Schematic illustration of cavitation enhances the delivery of ICG into the tumor cells *in vitro*. **(B)** The fluorescence images of MB49 cells incubated with ICG-GVs either with acoustic burst or without acoustic burst. Scale bar = 20 μm. **(C)** The fluorescence images and **(D)** quantification of MB49 cells after photothermal treatment. Viable cells were stained green with calcein-AM, and dead/later apoptosis cells were stained red with PI. Scale bar = 200 μm. Data are mean ± SD (*n* = 3 independent experiments). One-way ANOVA. $^{**}P < 0.01$, $^{****}P < 0.001$ *vs.* control. **(E)** Nonlinear contrast ultrasound images of GVs and ICG-GVs during tumor perfusion and reperfusion *in vivo*. **(F)** NIRF images of tumor-bearing mice at different time after injuction of ICG, ICG-GVs, or ICG-GVs + US burst, respectively. **(G)** Tissue distribution images of ICG in heart, liver, spleen, lung, kidney, and tumor at 12 h after injection of ICG, ICG-GVs, or ICG-GVs + US burst. Data are presented as mean ± SD (*n* = 3). One-way ANOVA. $^{**}P < 0.01$. **(H)** Time-course quantitative analysis of ICG fluorescence intensity in tumor tissues. Data are presented as mean ± SD (*n* = 3). One-way ANOVA. $^{**}P < 0.01$. The underlying numerical data for this figure can be found in S1 Data.

ICG-GVs + Laser group (Fig 3D), indicating acoustic delivery of ICG can significantly improve the PTT efficacy. No significant cytotoxicity was observed for ICG-GVs at the tested concentrations (S3 Fig).

We next evaluated the pharmacokinetic profile of ICG-GVs to assess its circulation stability *in vivo*. Healthy mice were intravenously injected with equal doses of free ICG or ICG-GVs (ICG dose: 2 mg/kg), and blood samples were collected at various time points post-injection for plasma ICG concentration measurement. As shown in S4 Fig, free ICG was rapidly cleared from circulation, falling below the detection limit within 4 min, with a half-life of approximately $3.2 \pm 0.5$ min. In contrast, ICG-GVs exhibited significantly prolonged circulation, reaching half of the initial concentration at 20 min post-injection, with an elimination half-life extended to $22.5 \pm 3.1$ min and an approximately 9-fold increase in $AUC_{0-24}$. These results demonstrate that covalent conjugation to GVs substantially improves the *in vivo* delivery stability of ICG.

To further explore the *in vivo* acoustic delivery effect, we intravenously injected ICG-GVs ($OD_{500} = 3.0$, ICG: 40 μg/mL) or free ICG, followed by acoustic irradiation. Fig 3E clearly revealed the tumor perfusion of ICG-GVs. The acoustic irradiation collapsed these GVs and re-perfusion could be observed, enabling repeated ultrasound irradiation of the tumor. Fig 3F shows representative NIRF images of tumors at various time points (0.5, 2, 4, 8, and 12 h) post-injection, demonstrating that burst acoustic irradiation significantly improved intratumoral delivery efficiency of ICG. Tissue distribution images at 12 h post-injection (Fig 3G) revealed that ICG was mainly accumulated in the liver and lung, but not in the tumor, for mice injected with free ICG or non-irradiated ICG-GVs. In contrast, upon burst acoustic irradiation, substantial ICG fluorescence was observed in the tumor of mice injected with ICG-GVs. The time-course fluorescence intensity curves in tumor tissues (Fig 3H) further confirmed that the ICG-GVs + US group exhibited significantly higher tumor accumulation at all time points compared to the non-irradiated ICG-GVs and free ICG groups.

## 3.4 The photothermal characterization of ICG-GVs

To explore the mechanisms in the *in vitro* tumor cell killing, we tested the photothermal performance of ICG-GVs. The given amount of ICG-GVs were added into the 96-well plates and were irradiated at 0.5, 0.75, 1.0, or 1.25 W laser power for 3 min. Fig 4A clearly showed that the temperature of samples at all of laser power intensities gradually increased with the increase of laser exposure time. The stronger the laser power was used, the higher temperature elevation would be. The 1.25 W laser irradiation had the most significant temperature elevation, achieving near 70 °C (Fig 4B). When keeping the laser power intensity at 0.75 W, increasing the concentrations of ICG-GVs would produce stronger temperature elevation (Fig 4C). For example, the temperature could achieve about 70 °C when the concentration of ICG in ICG-GVs was 40 μg/mL, while only 32 °C could achieve when the concentration of ICG was 10 μg/mL. In contrast, PBS control only showed hardly temperature elevation (<2 °C) (Fig 4C). Also, the photothermal effects of ICG-GVs could be effectively repeated with the switch of the 808 nm laser, showing their good stability (Fig 4D). These data indicated the strong tumor cell killing mainly contribute to heat-related mortality.

The enhanced ICG delivery observed upon ultrasound irradiation can be attributed to acoustic cavitation effects of GVs. Specifically, ultrasound-triggered GV cavitation induces transient pore formation in cell membranes (sonoporation) and generates microstreaming/microjets that temporarily disrupt vascular endothelial tight junctions, thereby increasing both cellular and vascular permeability [21,22]. Unlike conventional micrometer-sized microbubbles confined to blood vessels, the nanoscale dimensions (~200 nm) of GVs enable them to extravasate via the EPR effect and directly contact tumor cells. Upon ultrasound exposure, GVs release internal gas, coalesce into larger bubbles, and undergo inertial cavitation, producing localized mechanical forces that facilitate intracellular ICG delivery. This mechanistic explanation is consistent with our observations of significantly enhanced ICG fluorescence in ultrasound-irradiated tumor cells and tissues (Fig 3B and 3F–H). To further examine whether the acoustic delivery of ICG-GVs enhances tumor photothermal effects, the tumor-bearing mice were intravenously injected with ICG-GVs and then the tumor were received with or without acoustic irradiation, followed by laser irradiation. Fig 4E showed that the tumor temperature rose to 64.2 °C in the ICG-GVs + US+Laser group and kept the relatively stable under laser irradiation, suggesting that the acoustic delivery

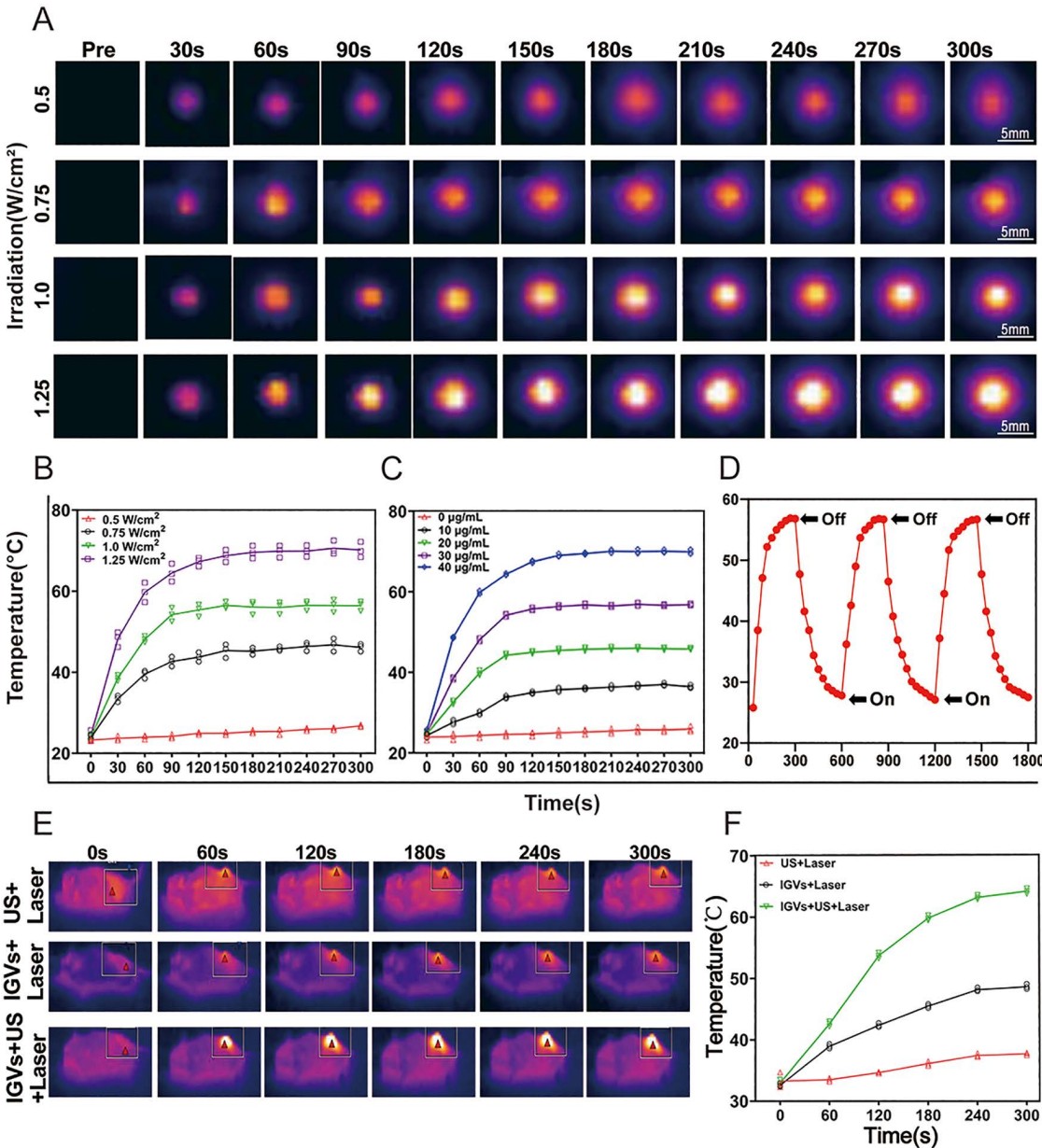

**Fig 4. The photothermal characterization of ICG-GVs. (A)** The temperature change profiles and (B) quantification of irradiated ICG-GVs. Data are mean ± SD (*n* = 3). **(C)** The temperature change profiles of irradiated ICG-GVs coated with different ICG concentration. Data are mean ± SD (*n* = 3). **(D)** The temperature change profiles of ICG-GVs with the repeated switch of the 808 nm laser. Data are mean ± SD (*n* = 3). **(E)** The temperature change profiles and (F) quantification of laser-irradiated tumor area after ICG-GVs injection with or without acoustic irradiation. Data are mean ± SD (*n* = 3). Unpaired *t* test. The underlying numerical data for this figure can be found in S1 Data.

promoted the ICG delivery into the tumor. By contrast, the tumor temperature of the mice receiving ICG-GVs without acoustic irradiation showed a slight increase with the peak temperature of 48.6 °C due to the relatively few ICG delivery into the tumor. No significant temperature elevation after the same laser irradiation was observed in the US+Laser control group (Fig 4F).

## 3.5 The *in vivo* anti-tumor efficacy

The *in vivo* acoustic delivery and photothermal effects of ICG-GVs motivate us to further explore the anti-tumor efficacy of ICG-GVs. To test it, we established the subcutaneous xenotransplanted tumor model through injecting the MB49 cells into the left hindlimb tumor model. When the tumor became palpable (about 100 mm³ in tumor volume), the tumor-bearing mice were randomly divided into six groups, including the control group, US + Laser group, ICG-GVs group, ICG-GVs + US group, ICG-GVs + Laser group and ICG-GVs + US + Laser group. The mice in the control group were intravenously administrated with 100 μL PBS and the mice in the US + Laser group were only received with acoustic irradiation and laser irradiation. The mice in the ICG-GVs group, ICG-GVs + US group, ICG-GVs + Laser group and ICG-GVs + US + Laser group were all intravenously administrated with ICG-GVs (OD$_{500}$ = 3.0 in 100 μL PBS), but the ICG-GVs + US group were only received with acoustic irradiation and ICG-GVs + Laser group only with laser irradiation. ICG-GVs + US + Laser group were treated with acoustic irradiation and laser irradiation (Fig 5A). From the Fig 5B, we can see that the US + Laser group and ICG-GVs + US group did not exhibit any tumor growth inhibition effect when compared with the control group ($P > 0.05$). The ICG-GVs + Laser group and ICG-GVs + US group showed partly tumor growth inhibition during 20 days. By contrast, the tumor growth of mice in the ICG-GVs + US + Laser group was significantly inhibited, achieving almost complete tumor regression (Fig 5B and 5C). A significant longer survival time was also observed for ICG-GVs + US + Laser group than these of the other groups (Fig 5D). No significant changes of body weight were found in all animals during the treatment process (Fig 5E).

To further confirm the anti-tumor effects of PTT, TUNEL, and H&E histological staining analysis for these tumors were performed. No apparent apoptotic cells and tumor necrosis could be seen in the control, US + Laser group, ICG-GVs group, ICG-GVs + US group, some apoptotic cells and part tumor necrosis could be observed in the ICG-GVs + Laser group. By contrast, significantly more apoptotic cells and obvious tumor necrosis could be observed in the tumors of ICG-GVs + US + Laser group (Figs 5B, 5E, and S8). The biosafety of ICG-GVs + US + Laser was also evaluated by H&E staining of the major organs (heart, liver, spleen, kidney, lung, and kidney). No histopathological changes were observed in the heart, liver, spleen, lung, and kidney, similar with the control group (S7 Fig). Furthermore, no significant differences were observed in hematology markers (S5 Fig) or in serum biochemical indicators of liver and kidney function (S6 Fig) between the control and ICG-GVs + US + Laser groups. All of these results indicated that ICG-GVs had good biosafety and could be used as PTT agents which might achieve visibly acoustic delivery of ICG for tumor photothermal treatment.

## 4. Discussion

PTT has emerged as an effective and promising cancer treatment modality due to its unique advantages, including noninvasiveness, ease of administration, and minimal toxicity to normal cells. To date, numerous PTT agents have been developed, yet few are suitable for clinical translation. ICG, the only FDA-approved photothermal agent, possesses inherent drawbacks such as poor water solubility, short blood half-life, and lack of tumor-targeted delivery. In this study, we developed a novel ICG-GVs platform that enables multimodal imaging (ultrasound, NIRF, and PA) and acoustically targeted delivery of ICG to tumors (Fig 6). The integrated imaging capability of ICG-GVs allows visualization of the entire delivery process—from tumor perfusion via ultrasound imaging to post-cavitation ICG distribution via NIRF and PA imaging [17,23,24]. More importantly, acoustic burst of ICG-GVs significantly enhances intratumoral ICG delivery, thereby improving the anti-tumor efficacy of PTT [25].

Unlike conventional phospholipid microbubbles (e.g., SonoVue) that lack reactive surface groups, biosynthetic GVs possess surface-accessible amino groups that enable stable amide bond formation with ICG, serving as the technical foundation for the "visibly acoustic delivery" achieved in this study. Our platform offers several distinct advantages over traditional microbubbles. First, covalent conjugation of ICG to GVs via amidation ensures stable binding, prolonging the blood half-life of ICG (S4 Fig). Second, the nanoscale size (~200 nm) of GVs provides a larger specific surface area compared to micrometer-sized microbubbles, allowing higher ICG loading per unit mass. Third and most importantly,

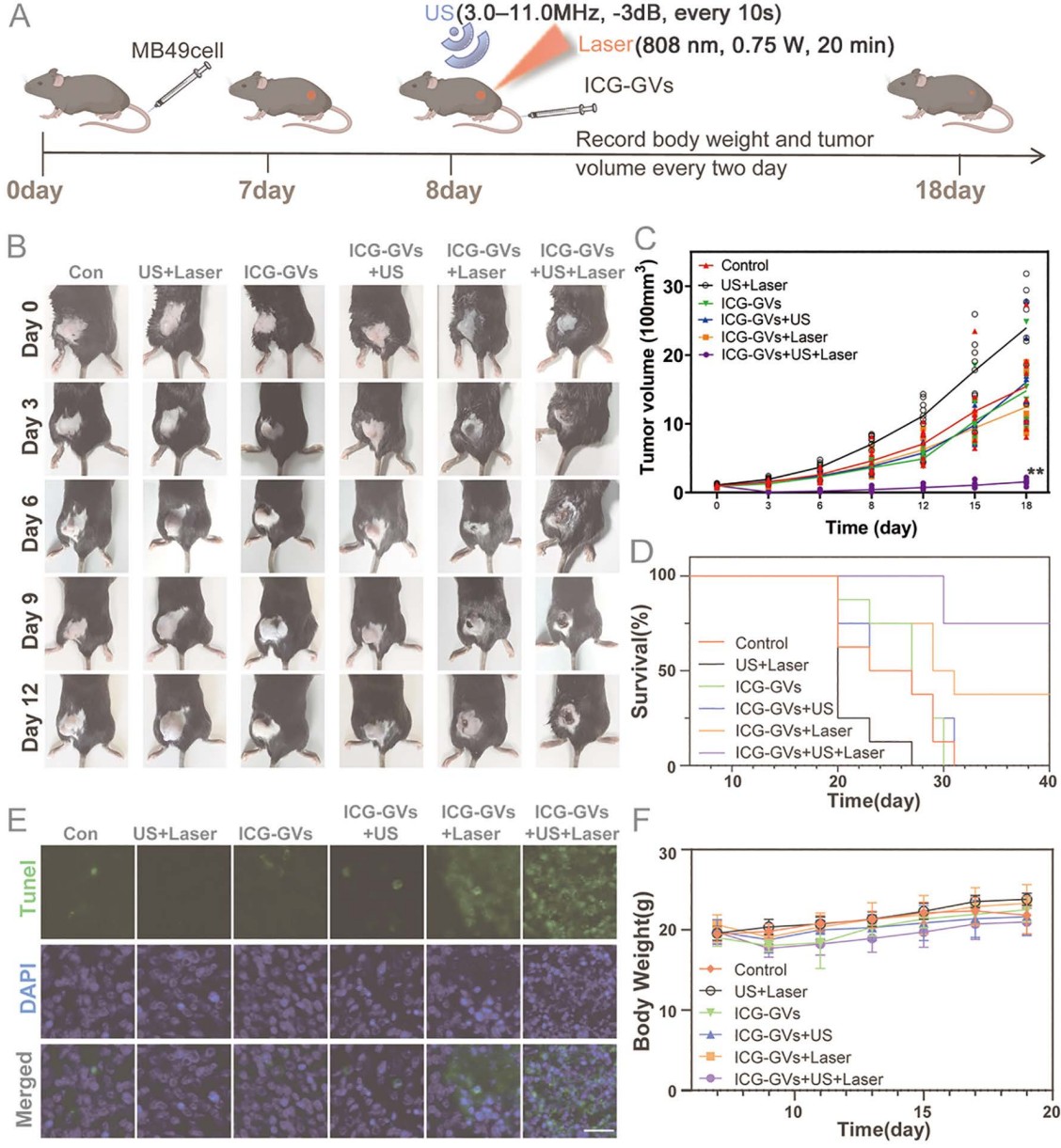

**Fig 5. The *in vivo* anti-tumor efficacy of visible acoustic ICG delivery via ICG-GVs therapy. (A)** Schematics of acoustic ICG delivery tumor photo-thermal therapeutic regimen. **(B)** Representative images of tumors in every group after treatments. **(C)** The tumor growth curves of mice received different treatments. Data are mean ± SD ($n = 8$ per group). Two-way ANOVA. ${}^{**}P < 0.01$ *vs.* control. **(D)** Survival curve of the tumor-bearing mice after different treatments. Log-rank test. **(E)** TUNEL staining of tumors after different treatment. Scale bar = 100 μm. **(F)** The body weight changes of mice after different treatments. Data are mean ± SD ($n = 8$ per group). Ultrasound parameters: 3.0–11.0 MHz, −3 dB power, bursts every 10 s until ICG-GVs disappeared. Laser parameters: 808 nm, 0.75 W, 20 min irradiation at 4 h post-injection.

the nanoscale dimensions of ICG-GVs facilitate extravasation through tumor vasculature via the enhanced permeability and retention effect, enabling direct contact with tumor cells. Upon ultrasound irradiation, these nanoscale GVs undergo cavitation, generating mechanical forces that transiently permeabilize tumor cell membranes and greatly enhance ICG delivery efficiency. In contrast, conventional microbubbles (1–5 μm) are confined within blood vessels, and any

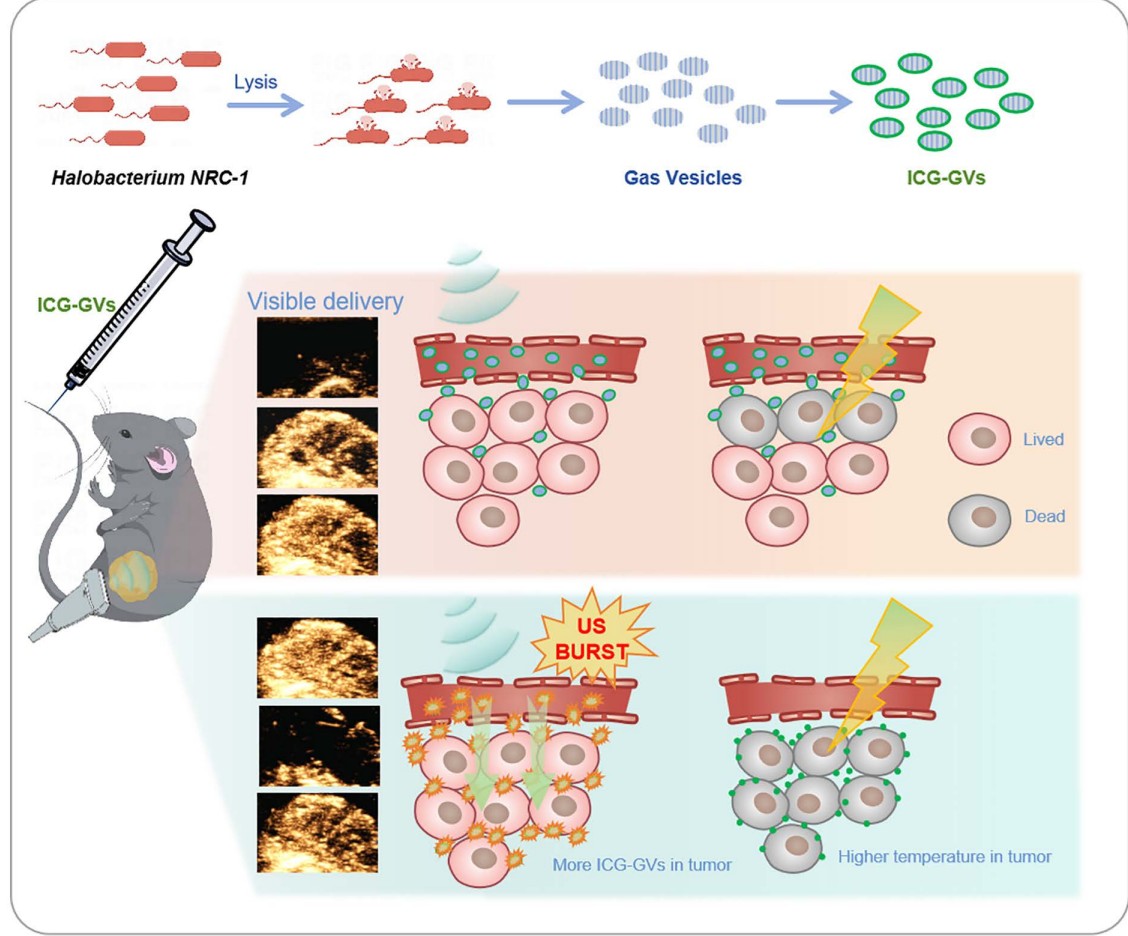

**Fig 6. Illustration of ICG-GVs for visible acoustic delivery of ICG in tumor for PTT.** GVs isolated from Halobacterium salinarum NRC-1 were modified by ICG to get ICG-GVs. ICG-GVs were locally delivered into the tumor by acoustic irradiation. The multimodality imaging capability of ICG-GVs visualized the whole ICG delivery processes. Acoustic burst of ICG-GVs significantly enhanced ICG delivery into the tumor, which greatly improved the anti-tumor outcome of PTT.

cavitation-induced effects occur primarily on vascular endothelial cells, substantially reducing ICG delivery to tumor cells. Indeed, previous studies have demonstrated that GVs interacting with ultrasound can effectively enhance biophysical effects including endocytosis, tight junction disruption, perivascular space expansion, and extracellular matrix penetration through cavitation [26].

Previous reports have established that PTT above 50 °C induces irreversible cell injury, membrane collapse, protein denaturation, and enzymatic inactivation, leading to coagulative necrosis, whereas temperatures between 42 and 45 °C result in sublethal and reversible cellular responses [27,28]. Higher photothermal temperatures therefore produce stronger anti-tumor efficacy and reduce the risk of tumor recurrence. In our study, ICG-GVs + US + Laser treatment achieved tumor temperatures exceeding 60°C, causing irreversible damage to tumor cells. This pronounced thermal effect is attributed to the efficient acoustic delivery of ICG into the tumor tissue.

We acknowledge certain limitations of the current study, particularly regarding the tumor model employed. While subcutaneous xenograft models are widely used for initial proof-of-concept evaluation, they do not fully recapitulate the complexity of orthotopic liver tumors, particularly with respect to the tumor microenvironment and anatomical challenges

associated with deep-seated organs. An orthotopic liver cancer model would indeed provide a more rigorous assessment of tumor targeting and therapeutic efficacy. However, two major factors currently preclude such experiments within the scope of this work. First, the inherent pharmacokinetic properties of ICG lead to pronounced nonspecific accumulation in normal liver tissue, a well-documented phenomenon that complicates the interpretation of targeted delivery in orthotopic settings [29,30]. This hepatic background uptake is an intrinsic characteristic of ICG rather than a limitation of our GVs-based delivery system. Second, the therapeutic efficacy of PTT is fundamentally constrained by the limited tissue penetration depth of light in the near-infrared I (NIR-I) window (700–900 nm) employed in this study [31,32]. Although ultrasound can penetrate deeply to trigger cavitation-mediated delivery, the 808 nm laser required for ICG excitation cannot effectively reach tumors located deep within the liver parenchyma without significant attenuation and potential thermal damage to overlying tissues [33]. Emerging strategies utilizing the second near-infrared (NIR-II) window (1,000–1,700 nm) offer improved tissue penetration and reduced scattering [34], but their integration with ICG-based platforms would require substantial redesign beyond the current study's scope.

Therefore, while acknowledging the value of orthotopic models, we opted for a subcutaneous model to unequivocally validate the core mechanism of acoustically enhanced ICG delivery and its photothermal efficacy under well-controlled conditions. The ultrasound parameters employed (−3 dB power, 10 s interval), validated in our previous work, proved sufficient to demonstrate this core concept; however, systematic optimization of parameters (e.g., mechanical index, pulse length) may further enhance delivery efficiency and represents a key focus of our subsequent investigations. From a translational perspective, GVs can be produced at scale via microbial fermentation, and the ICG conjugation and purification processes are straightforward and controllable. Preliminary stability data support favorable storage stability, and future studies will systematically evaluate critical parameters such as batch-to-batch consistency. Future investigations will also focus on adapting this acoustic delivery strategy to NIR-II photothermal agents and evaluating its performance in orthotopic liver tumor models, aiming to translate the demonstrated benefits into more clinically relevant settings.

## 5. Conclusions

In summary, we developed nanoscale biosynthetic ICG-GVs that enable remotely controlled, visibly acoustic delivery of ICG to tumors. The integrated trimodal imaging capability (ultrasound, NIRF, and PA) allows real-time visualization of the delivery process, while ultrasound-triggered GV cavitation substantially enhances intratumoral ICG accumulation. This platform significantly prolongs ICG circulation, achieves effective photothermal ablation with elevated tumor temperatures, and promotes complete tumor regression with excellent biosafety. Our study presents a clinically translatable strategy for precise and effective ICG-based PTT.

## Supporting information

**S1 Data. Excel file containing the underlying numerical data for all figures in the main manuscript and supporting information.** Each sheet corresponds to a specific figure.
(XLSX)

**S1 Materials and Methods. Pharmacokinetic analysis.**
(DOCX)

**S1 Fig. Isolation and purification of GVs.** Schematic illustration of the procedure for isolation and purification of GVs from Halobacterium NRC-1. **(A)** Initial culture system. **(B)** Mature bacterial liquid system after cultivation. **(C)** Bacterial liquid transferred into a separatory funnel awaiting collection. **(D)** GVs obtained after the first centrifugation. **(E)** GVs obtained after the final centrifugation. **(F)** GVs collected together.
(PDF)

**S2 Fig. Long-term stability of ICG-GVs.** The stability of ICG-GVs was monitored in PBS over 7 days. Data are presented as mean ± SD. Each dot represents an individual measurement ($n = 3$ per group). The underlying numerical data for this figure can be found in S1 Data.
(PDF)

**S3 Fig. Biocompatibility of ICG-GVs.** Assessment of MB49 cell viability following a 48-hour exposure to ICG-GVs (10–40 µg/mL). Data are presented as mean ± SD. Each dot represents an individual measurement ($n = 3$ per group). The underlying numerical data for this figure can be found in S1 Data.
(PDF)

**S4 Fig. Pharmacokinetic analysis of ICG-GVs and free ICG in healthy mice.** Plasma concentration-time profiles of ICG after intravenous injection of free ICG or ICG-GVs (ICG dose: 2 mg/kg) in healthy C57BL/6 mice. Blood samples were collected at indicated time points and ICG concentration was measured by fluorescence spectrophotometry. Data are presented as mean ± SD. Each dot represents an individual measurement ($n = 3$ per group). The underlying numerical data for this figure can be found in S1 Data.
(PDF)

**S5 Fig. ICG-GVs+US+Laser treatment had no hematological toxicity.** Hematology profiles, **(A)** Hematocrit (HCT), **(B)** white blood cell count (WBC), **(C)** platelet count (PLT), **(D)** hemoglobin (HGB), **(E)** red blood cell count (RBC), **(F)** mean corpuscular hemoglobin (MCH), **(G)** mean corpuscular volume (MCV), and **(H)** mean corpuscular hemoglobin concentration (MCHC), show no significant differences between the control and the ICG-GVs + US + Laser treatment groups. Data are presented as mean ± SD. Each dot represents an individual measurement ($n = 3$ per group). The underlying numerical data for this figure can be found in S1 Data.
(PDF)

**S6 Fig. Liver and kidney function parameters in control and treated groups. (A)** ALT, **(B)** AST, **(C)** BUN, and **(D)** CREA levels in control and ICG-GVs + US+Laser treated groups. Data are presented as mean ± SD. Each dot represents an individual measurement ($n = 3$ per group). The underlying numerical data for this figure can be found in S1 Data.
(PDF)

**S7 Fig. Absence of organ damage.** H&E staining of major organs (heart, liver, spleen, lung, kidney) in control and ICG-GVs group. Scale bar = 200 µm.
(PDF)

**S8 Fig. Histological staining of seed tumor necrosis.** H&E staining of tumors after different treatment. Scale bar = 200 µm.
(PDF)

## Author contributions

**Data curation:** Licong Huang, Chenhui Li.

**Formal analysis:** Shuhui Wang.

**Methodology:** Jiaqi Zhang, Jingwen Ding, Chenhui Li.

**Software:** Licong Huang, Yuping Yang.

**Supervision:** Ping Zhao, Fei Yan.

**Writing – original draft:** Jiaqi Zhang, Shuhui Wang.

**Writing – review & editing:** Ping Zhao, Qian Li, Fei Yan.

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
