## [Editor Report · Decision Letter 0]

5 Dec 2025

Dear Dr Li,

Thank you for submitting your manuscript entitled "Visibly acoustic delivery of ICG via biosynthetic gas vesicles for tumor photothermal therapy" for consideration as a Research Article by PLOS Biology. Please accept my sincere apologies for the delay in getting back to you with feedback.

Your manuscript has now been evaluated by the PLOS Biology editorial staff and I am writing to let you know that we would like to send your submission out for external peer review.

Once your full submission is complete, your paper will undergo a series of checks in preparation for peer review. After your manuscript has passed the checks it will be sent out for review. To provide the metadata for your submission, please Login to Editorial Manager (https://www.editorialmanager.com/pbiology) within two working days, i.e. by Dec 07 2025 11:59PM.

Kind regards,

Richard

Richard Hodge, PhD

rhodge@plos.org

PLOS

---

## [Decision Letter · Decision Letter 1]

28 Jan 2026

Dear Dr Li,

Thank you for your patience while your manuscript "Visibly acoustic delivery of ICG via biosynthetic gas vesicles for tumor photothermal therapy" was peer-reviewed at PLOS Biology as a Research Article. Please accept my sincere apologies for the delays that you have experienced during the peer review process. Your manuscript has been evaluated by the PLOS Biology editors, an Academic Editor with relevant expertise, and by two independent reviewers.

As you will see in the reviewer reports, which can be found at the end of this email, although the reviewers find the work potentially interesting, they have also raised a substantial number of important concerns. Based on their specific comments and following discussion with the Academic Editor, it is clear that a substantial amount of work would be required to meet the criteria for publication in PLOS Biology. However, given our and the reviewer interest in your study, we would be open to inviting a comprehensive revision of the study that thoroughly addresses all the reviewers' comments. Given the extent of revision that would be needed, we cannot make a decision about publication until we have seen the revised manuscript and your response to the reviewers' comments. Your revised manuscript would need to be seen by the reviewers again, but please note that we would not engage them unless their main concerns have been addressed.

Specifically, the reviewers agree that the delivery approach is interesting, but they raise some partly overlapping concerns. Both reviewers note that the manuscript lacks direct comparative data to either clinically used ultrasound microbubbles or free ICG to demonstrate a proven advantage in performance. In addition, Reviewer #2 raises concerns that important in vivo safety assessments are missing and that the therapeutic relevance of the study would be elevated by including an orthotopic liver cancer model to demonstrate tumor targeting.

We appreciate that these requests represent a great deal of extra work, and we are willing to relax our standard revision time to allow you 6 months to revise your study. Please email us (plosbiology@plos.org) if you have any questions or concerns, or envision needing a (short) extension.

**IMPORTANT - SUBMITTING YOUR REVISION**

*Resubmission Checklist*

*Published Peer Review*

*PLOS Data Policy*

*Blot and Gel Data Policy*

Kind regards,

Richard

Richard Hodge, PhD

rhodge@plos.org

REVIEWS:

Reviewer #1: The authors present a biosynthetic gas vesicle (GV)-based delivery system for indocyanine green (ICG) that addresses several intrinsic limitations of the only FDA-approved photothermal agent, including poor aqueous stability, a short in vivo half-life, and insufficient tumor-targeted accumulation. In this system, ICG is covalently conjugated to the surface of GVs via amide bonds, yielding a nanoscale probe with an average size of approximately 264 nm. The resulting ICG-GVs enable multimodal imaging, including ultrasound, near-infrared fluorescence, and photoacoustic imaging, allowing real-time visualization of ICG delivery to tumors. Importantly, ultrasound-induced cavitation enhances cellular uptake of ICG, and when combined with 808 nm laser irradiation, the platform achieves an in vitro tumor cell killing rate exceeding 90%. In vivo, tumor temperatures reach up to 64.2 °C, leading to effective tumor regression and prolonged survival in tumor-bearing mice. The system exhibits favorable biosafety, with no significant organ toxicity, highlighting its potential as a controllable, precise, and clinically translatable strategy for ICG-mediated photothermal therapy.

Despite these promising results, several issues should be addressed to further strengthen the manuscript and enhance its translational value.

1. The authors do not clearly compare ICG-GVs with clinically used ultrasound microbubbles in terms of ICG loading efficiency, in vivo circulation half-life, and tumor-targeting performance. Such comparisons are essential to demonstrate the unique advantages of GVs.

2. Key parameters relevant to clinical translation, including large-scale production feasibility, batch-to-batch consistency, and storage stability, are not discussed.

3. Although ICG is covalently coupled via amide bonds, the manuscript does not report quantitative values for ICG loading capacity or encapsulation efficiency.

4. Only a single ultrasound condition (−3 dB power, 10 s interval) is used throughout the study, without justification or optimization.

5. The scale bar in the TEM image (Figure 2C) is listed as 200 nm but is not clearly labeled. In addition, the grouping information in the fluorescence images (Figures 4B and 4C) is ambiguous, particularly regarding the presence or absence of ultrasound irradiation.

6. The tumor growth curve in Figure 6C displays only mean values without individual data points, obscuring data variability. Overlaying individual measurements would improve transparency.

7. The "Key Findings" sections in Tables 1 and 2 are overly verbose and partially redundant with the main text.

8. The treatment scheme in Figure 6A does not specify key ultrasound (power, duration, interval) and laser (wavelength, power density, irradiation time) parameters. Likewise, the quantitative imaging data in Figure 3D-F do not clearly define the measured indicators (for example, fluorescence intensity, signal-to-noise ratio).

9. Several abbreviations, including CEUS, PA, and PTT, are not defined at their first appearance in the figure captions, which does not conform to standard academic practice.

10 Most figure captions do not report sample sizes (n values) or statistical analysis methods, relying instead on descriptions in the main text.

11. The reference format needs to be unified, and some important references are missing, such as "Ultrasound-controlled drug release and drug activation for cancer therapy" and "Recent advances in indocyanine green-based probes for second near-infrared fluorescence imaging and therapy." Ensure that the reference format is consistent and check for any missing important references.

Reviewer #2: In this manuscript, Zhang et al. developed biosynthetic gas vesicles coated with ICG, which are designed to enhance and visualize the delivery of ICG and improve its anti-tumor efficacy in PTT. While the study presents an interesting approach, several serious concerns must be raised regarding the unclear scientific premise and significant weaknesses in the animal experiments described.

Major critics:

1. The manuscript claims that acoustic cavitation enhances ICG delivery into tumor cells, while the discussion of its underlying mechanism is inadequate. Please discuss the potential mechanism.

2. Regarding the pharmacokinetic evidence, the manuscript claims improved delivery stability for ICG-GVs, but a critical piece of evidence is missing: a direct comparison of the in vivo half-life between ICG-GVs and free ICG.

3. It is necessary to update both the manuscript and the figures to clearly specify the NIR and ultrasound parameters and the injection regimen of ICG-GVs in vivo anti-tumor therapy.

4. The clinical relevance of this study hinges on the safety of intravenous ICG administration. It is essential to include in vivo safety assessments, such as serum biochemistry, complete blood counts, and histopathological analysis of major organs, to rule out any adverse toxic effects.\

5. The biodistribution data shows predominant accumulation of ICG-GVs in the liver following intravenous administration. To more rigorously evaluate the translational potential for treating deep-seated tumors, it is strongly recommended to include an orthotopic liver cancer model in the study.

Minor issues:

1. The color scale bars in the figures should be revised. Please replace the non-quantitative labels "High" and "Low" with specific numerical values. Additionally, scale bars are missing in several figures (Fig. 2B, Fig. 3H, Fig. 3I, Fig. 4F, Fig. 4G, Fig. 5A).

2. Some of the presented data lack appropriate statistical analysis. Please add the necessary statistical analysis and specify the methods used (e.g., unpaired t-test, one-way ANOVA).

---

## [Decision Letter · Decision Letter 2]

9 Apr 2026

Dear Dr Li,

Thank you for your patience while we considered your revised manuscript "Visibly acoustic delivery of ICG via biosynthetic gas vesicles for tumor photothermal therapy" for publication as a Research Article at PLOS Biology. Please accept my sincere apologies for the delays that you have experienced during this round of the peer review process. This revised version of your manuscript has been evaluated by the PLOS Biology editors, the Academic Editor and the original reviewers.

Based on the reviews, I am pleased to say that we are likely to accept this manuscript for publication, provided you satisfactorily address the following data and other policy-related requests that I have provided below (A-H):

(A) We routinely suggest changes to titles to ensure maximum accessibility for a broad, non-specialist readership. In this case, we would suggest a minor edit to the title, as follows, since the general reader may be unclear about the term 'visibly acoustic'. Please ensure you change both the manuscript file and the online submission system, as they need to match for final acceptance:

“Acoustic delivery of indocyanine green via biosynthetic gas vesicles for tumor photothermal therapy”

(B) In the Abstract, please spell out the following abbreviations which are currently undefined - PTT, ICG, FDA. In addition, please specify that the FDA is the American FDA.

(C) You may be aware of the PLOS Data Policy, which requires that all data be made available without restriction: http://journals.plos.org/plosbiology/s/data-availability. For more information, please also see this editorial: http://dx.doi.org/10.1371/journal.pbio.1001797

-Supplementary files (e.g., excel). Please ensure that all data files are uploaded as 'Supporting Information' and are invariably referred to (in the manuscript, figure legends, and the Description field when uploading your files) using the following format verbatim: S1 Data, S2 Data, etc. Multiple panels of a single or even several figures can be included as multiple sheets in one excel file that is saved using exactly the following convention: S1_Data.xlsx (using an underscore).

-Deposition in a publicly available repository. Please also provide the accession code or a reviewer link so that we may view your data before publication.

Figure 2D-G, 3D-F, 4D, 4H, 5B-D, 5F, S2, S3, S4, S5A-H, S6A-D

NOTE: the numerical data provided should include all replicates AND the way in which the plotted mean and errors were derived (it should not present only the mean/average values)

(D) Please also ensure that each of the relevant figure legends in your manuscript include information on *WHERE THE UNDERLYING DATA CAN BE FOUND*, and ensure your supplemental data file/s has a legend.

(E) Per journal policy, if you have generated any custom code during the course of this investigation, please make it available without restrictions. Please ensure that the code is sufficiently well documented and reusable, and that your Data Statement in the Editorial Manager submission system accurately describes where your code can be found. More information on our Code Policy, what and how to share can be found here: https://journals.plos.org/plosbiology/s/code-availability

(F) Please ensure that your Data Statement in the submission system accurately describes where your data can be found and is in final format, as it will be published as written there.

(G) Please ensure that you are using best practice for statistical reporting and data presentation. These are our guidelines https://journals.plos.org/plosbiology/s/best-practices-in-research-reporting#loc-statistical-reporting and a useful resource on data presentation https://journals.plos.org/plosbiology/article?id=10.1371/journal.pbio.1002128

- If you are reporting experiments where n ≤ 5, please plot each individual data point.

(H) Please note that per journal policy, the model system/species studied should be clearly stated in the abstract of your manuscript.

We expect to receive your revised manuscript within two weeks.

*Published Peer Review History*

*Press*

Best regards,

Richard

Richard Hodge, PhD

rhodge@plos.org

Reviewer remarks:

Reviewer #1: The revised manuscript has satisfactorily addressed all of my comments and concerns. I therefore recommend that the manuscript be accepted for publication in PLOS Biology without further revision.

Reviewer #2: They have addressed most of my concerns. I have no more questions.

---

## [Editor Report · Decision Letter 3]

20 Apr 2026

Dear Dr Li,

On behalf of my colleagues and the Academic Editor, Baojun Wang, I am pleased to say that we can accept your manuscript for publication, provided you address any remaining formatting and reporting issues. These will be detailed in an email you should receive within 2-3 business days from our colleagues in the journal operations team; no action is required from you until then. Please note that we will not be able to formally accept your manuscript and schedule it for publication until you have completed any requested changes.

PRESS

Best wishes,

Richard

Richard Hodge, PhD

rhodge@plos.org

PLOS
